**Subject Category:**
Biology (whole organism)

behaviour/ecology/evolution

predation risk, predator–prey interaction, experiment, colour-polymorphism, background crypsis, visual acuity

**Author for correspondence:**
Carina Nebel
e-mail: carina.nebel@gmail.com

# Response time of an avian prey to a simulated hawk attack is slower in darker conditions, but is independent of hawk colour morph

Carina Nebel[1], Petra Sumasgutner[1], Adrien Pajot[1,2] and Arjun Amar[1]

[1]FitzPatrick Institute of African Ornithology, DST-NRF Centre of Excellence, University of Cape Town, Private Bag X3, Rondebosch, 7701 Cape Town, South Africa
[2]Bordeaux Sciences Agro, 1 Cours du Général de Gaulle, 33170 Gradignan, France

CN, 0000-0002-0848-1676; PS, 0000-0001-7042-3461;
AP, 0000-0002-6874-4490; AA, 0000-0002-7405-1180

To avoid predation, many species rely on vision to detect predators and initiate an escape response. The ability to detect predators may be lower in darker light conditions or with darker backgrounds. For birds, however, this has never been experimentally tested. We test the hypothesis that the response time of avian prey (feral pigeon *Columbia livia f. domestica*) to a simulated hawk attack (taxidermy mounted colour-polymorphic black sparrowhawk *Accipiter melanoleucus*) will differ depending on light levels or background colour. We predict that response will be slower under darker conditions, which would translate into higher predation risk. The speed of response of prey in relation to light level or background colour may also interact with the colour of the predator, and this idea underpins a key hypothesis proposed for the maintenance of different colour morphs in polymorphic raptors. We therefore test whether the speed of reaction is influenced by the morph of the hawk (dark/light) in combination with light conditions (dull/bright), or background colours (black/white). We predict slowest responses to morphs under conditions that less contrast with the plumage of the hawk (e.g. light morph under bright light or white background). In support of our first hypothesis, pigeons reacted slower under duller light and with a black background. However, we found no support for the second hypothesis, with response times observed between the hawk-morphs being irrespective of light levels or background colour. Our findings experimentally confirm that birds detect avian predators less efficiently under darker conditions. These

conditions, for example, might occur during early mornings or in dense forests, which could lead to changes in anti-predator behaviours. However, our results provide no support that different morphs may be maintained in a population due to differential selective advantages linked to improved hunting efficiencies in different conditions due to crypsis.

## 1. Introduction

Predation is a strong evolutionary force that will shape prey behaviour and consequently population dynamics [1–3]. The spatial and temporal distribution of prey in the landscape may be a function of both the distribution of resource and of predators [4–8]. A prey's ability to detect a predator and decide on an appropriate response are important factors that will influence predation risk [2,9]. Many diurnal species rely primarily on vision to detect predators. This may be compromised by environmental factors, such as lower ambient light levels and the nature of the background cover [1,10–15]. Birds, as vision-dependent species, are known to reduce activity in conditions where they are less able to detect predators [11,12,16–18].

Some habitats might also offer greater concealment for predators of varying phenotypes. Differences in concealment and detectability due to crypsis have been hypothesized for the maintenance of colour-polymorphism in raptors [19–21]. Barn owl females (*Tyto alba*) show different habitat use, with reddish females occupying territories with less wooded areas compared to white females. In the tawny owl (*Strix aluco*), rufous birds occupied more wooded territories than grey birds, a pattern thought to be driven by crypsis advantages for the different morphs [22]. The hypothesis for a selective advantage of certain morphs under different environmental conditions has recently received some empirical support in the colour-polymorphic black sparrowhawk (*Accipiter melanoleucus*) [23,24]. One of these studies found that nestling prey provisioning rates by the different morphs were dependent on light levels, i.e. light morphs provisioned more in brighter, and dark morphs provisioned more in duller light conditions [24]. The mechanism behind this finding was also hypothesized to be linked to improved crypsis of the two morphs under different environmental conditions, resulting in higher hunting success and thus higher provisioning rates [24]. However, other data from the black sparrowhawk study system provided less support for this idea, whereby dark morph black sparrowhawks forage more under dull light condition while light morphs did not show a light-dependent activity pattern [25]. This suggests that a dark morph bird's higher provisioning under dull light conditions might simply reflect increased foraging effort during these conditions, rather than any improvement in foraging success. However, it does also suggest that light morphs may be more successful at hunting in brighter conditions. The mechanism for any such cryptic advantage for light morph individuals under bright light conditions remains as yet untested.

In this study, we experimentally test the hypothesis that reaction time of an avian prey (feral pigeon, *Columba livia f. domestica*) to a simulated attacking avian predator (a taxidermy mounted black sparrowhawk) will differ depending on light conditions or background coloration, with the prediction being that speed of response will be slower under dull light conditions or with a darker background. Additionally, we test the hypothesis that response time will differ depending on the morph of the attacking hawk in relation to either light condition or background colour, with our prediction being that response time will be slowest where the hawk's plumage contrasts less with light levels or background colour (i.e. a light morph under bright light conditions or with a white background).

## 2. Material and methods

The experimental trials were carried out in April and May (*control-hawk* and *light level-morph* experiments) and June 2018 (*background-morph* experiment, a fifth trial was used for the *light level-morph* experiment). In total, we caught 185 feral pigeons (hereafter: pigeons) on the Cape Peninsula, South Africa, which were then kept overnight at the experimental site in individual $50 \times 30 \times 30$ cm cages. While in our care (for a maximum of 24 h), pigeons had access to water *ad libitum* but were only fed during the experimental trials the next day.

We used four male black sparrowhawks (hereafter: hawk), with two replicates per morph, which were taxidermy mounted in a flight position (electronic supplementary material, figure S1). They had the same wingspan (67–72 cm), body length (45–48 cm) and weight (250 g). The black sparrowhawks

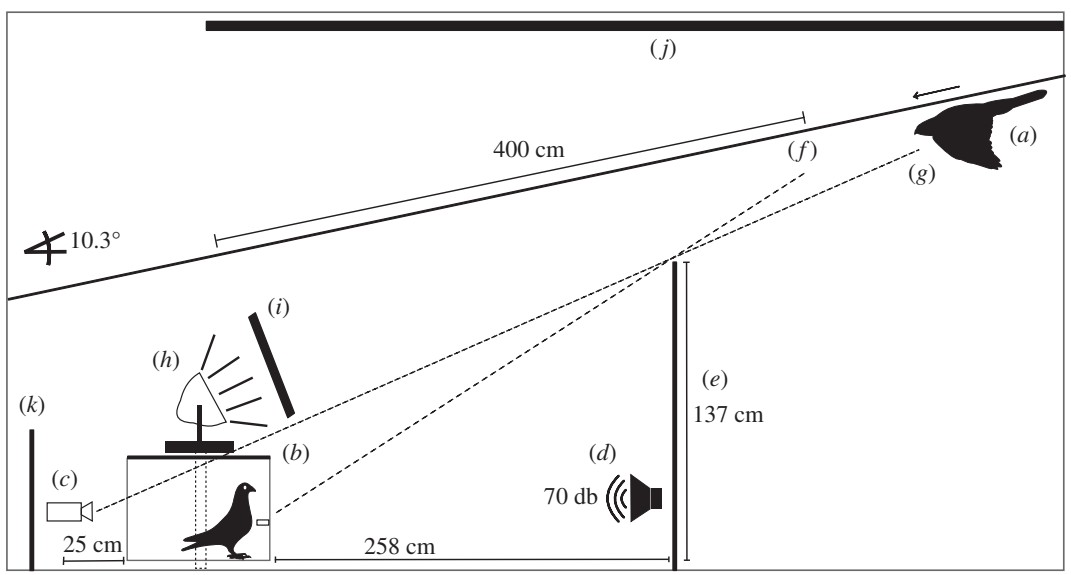

**Figure 1.** The experimental set-up. The hawk (*a*) was first visible to the pigeon (*b*) at point (*f*) and to the camera (*c*) at point (*g*). First visibility was standardized by a blind (*e*). Lamps (*h*) were dimmed with a black fabric for dull light conditions (*i*). Background colour was changed by spanning a black fabric above the set-up or using the white wall (*j*). We played white noise (*d*) and a second blind (*k*) was installed to block out the hawk to the pigeon after the attack.

were previously collected during the monitoring of the study population on the Cape Peninsula, South Africa, with causes of death being unknown in all cases. Black sparrowhawks regularly prey on feral pigeons in our study area and analysis of prey remains have shown that about 28% of all prey items recorded are feral pigeons [26].

To confirm that within our experimental set-up pigeons were able to recognize a predatory attack as such and did not only respond to a moving object *per se*, we used a control which was a squared fabric bag (electronic supplementary material, figure S2).

The experimental set-up was similar to other research [27–29] (figure 1) and consisted of a hawk 'flying' down a 12 m long and 10.3° angled line. A 137 × 120 m blind standardized the point of first visibility to the pigeon. The cage, containing a single pigeon, was placed in a fixed location, 1 m beneath the line. Food was provided from a feeder at the front of the cage thereby standardizing the pigeon's position during each trial.

The experiment was set up indoors (room: 15 × 4 × 2.9 m) with a light source consisting of four lamps (Lightstar, Professional Lighting), angled to shed light on the on-coming hawk (figure 1). The bulbs used were tungsten halogen incandescent lamps (Osram, 800 W, 240 V) that produce a continuous spectrum of light, including near UV light.[1] Our choice of positioning and angling of the lights was taken for several reasons: (i) to ensure that the lights did not result in direct glare to the pigeon that would have hindered its visual capabilities in detecting the hawk, (ii) to ensure that the ventral area of the hawk was illuminated in a manner that made it most easy to distinguish the two morphs, which differ in coloration on their throat, breast, belly and underwing coverts [30]. Alternatively placing the light behind the attacking hawk resulted in a backlit hawk silhouette that would have probably reduced the ability for the pigeon to differentiate between the two morphs. (iii) Positioning the light on either or each side of the experimental set-up resulted in the moving hawk casting a shadow of the hawk onto the wall that was visible to the pigeon before the actual hawk. Currently, almost nothing is known about how avian predators approach prey in relation to the sun, but based on the aforementioned considerations, we chose the present set-up as the most reasonable design to explore our hypotheses.

A high-speed camera (XDV 4 K camera, 90 fps) was mounted in a fixed position to record time of first visibility of the hawk and response of the pigeon. Food for the pigeon (a seed mixture for wild garden birds) was placed in a specific food hopper at the front of the cage. After initiating feeding, we waited 1 min before releasing the hawk, to prevent the pigeon from associating food with an attack. In the

---

[1]Zeiss-campus.magnet.fsu.edu: http://zeiss-campus.magnet.fsu.edu/articles/lightsources/tungstenhalogen.html (accessed 1 July 2019, 10:00).

case of an unsuccessful trial (i.e. the pigeon would not feed), we aborted the trial after 10 min and repeated after a minimum 20 min break. 20 min was also the minimum time between every trial. Pigeons were exposed to a maximum of five trials, all conducted on the same day.

We created two light treatments: 'bright light', using four lamps on highest intensity ($2182 \pm 65$ lx); and 'dull light' with two dimmed lamps ($112 \pm 12$ lx, figure 1). In comparison to a real-life situation, 112 lx would be comparable to the light during a very dark overcast day, i.e. as it is encountered in a thick forest or during the early or late hours of the day. The bright light situation, while considerably brighter, is comparable to an overcast day during noon. This maximum light intensity was limited by the luminance output of our lamps and the heat production. Our dullest light intensity was limited by the capabilities of our camera to record interpretable images (electronic supplementary material, table S1). Lux levels were measured at the blind with a digital multimeter (MS8229, Mastech). The same device was used to measure temperature (°C) at the pigeon cage after every trial. Between the first visibility of the hawk and when it passed over the pigeon, the mount covered a distance of 4 m. We played white noise throughout the trials to avoid any sound of the moving hawk mount (70 dB white noise, measured at the pigeon cage).

The *light level-morph* experiment consisted of a complete crossed design with four treatments (dark morph-dull light, dark morph-bright light, light morph-dull light, light morph-bright light) in randomized order. In all these trials, the background behind the hawk was the white colour of the ceiling. Throughout these trials, we also randomly substituted the hawk for the control to confirm that pigeons recognized the hawk as a predator (by reacting quicker; *control–hawk* experiment).

The *background-morph* experiment consisted of a complete crossed design (dark morph-black background, dark morph-white background, light morph-black background, light morph-white background). Background colour was changed by spanning a black fabric across the ceiling (figure 1). These experiments were all done under dull light conditions, as we did not have the capacity to conduct these treatments under both dull and bright light levels. No control object was used in this second set-up, but we added a random hawk-morph under bright light to increase the sample size of the *light level-morph* experiment and to have five trials per pigeon. We confirmed that the different hawk morphs were contrasting with the background colour by calculating the contrast ratio (relative luminance). For full details, see electronic supplementary material, table S2 and figures S3 and S4.

For each trial, we use the high-speed video to measure the duration from the moment the hawk came into view until two different responses by the pigeon: (a) detection time, the initiation of a head movement to face the attacking hawk; and (b) reaction time, the initiation of a physical escape response (in most cases, the pigeon would visibly tense and lower its body or escape into the back of the cage). Both measurements represent different cognitive processes: detection time is the detection of an on-coming object, whereas the reaction time will incorporate the detection, the perception of a predatory attack and the decision to respond. We were unable to measure reaction time in 9.5% of all trials, because the pigeon was already looking directly in the direction of the on-coming hawk. Additionally, we removed trials from the experiment where no measurable escape reaction was initiated (8.8% of all trials, because the pigeon did not show a measurable reaction). Time stamps were recorded on the high-speed video in slow motion with the free software MPC-HC 1.7.13.[2] Additionally, we measured the speed of the hawk and control by dividing the distance from the point of view to a fixed point towards the end of the line by time (m s$^{-1}$).

## 2.1. Statistical analysis

We fitted three different linear mixed models (LMM) using the 'lme4' package (v. 1.1–17; [31]) in the software R, version 1.1.442 [32] (R Core Team 2018). Our response variable was either (a) detection or (b) reaction time (both log-transformed), with the random term 'pigeon ID' fitted to control for lack of independence among trials conducted on the same pigeon. In all models, we tested for an influence of (i) speed of the hawk or control, (ii) pigeon's head position (head down at the feeder or looking up), (iii) temperature (°C), (iv) hawk replicate (1–4), (v) time of day (hour), (vi) number of trial (1–5) and (vii) number of experimental set-up (1–3; the experiment was set up in total three times). Final covariates were chosen by stepwise backwards elimination [33], until only terms that were significant at $p < 0.1$ level remained.

Model 1 (*control–hawk* experiment) tested whether response times differed between a hawk and a control; the key explanatory variable was mount type (hawk or control). Model 2 (*light level-morph*

[2]MPC-HC 1.7.13 can be obtained here: https://mpc-hc.org/ (13 February 2019).

experiment) explored whether response times changed under different light conditions, and whether this varied depending on the hawk-morph. In this model, we used all data with a hawk and a white background. Our key explanatory variables were light level (dull or bright light), hawk-morph (dark or light morph) and their interaction. Model 3 (*background-morph* experiment) explored whether response times differed between differently coloured backgrounds, and whether this varied depending on the hawk-morph. Our key variables were background colour (white or black background), hawk-morph (dark or light morph) and their interaction.

## 3. Results

Pigeons detected and reacted slower to the control than to a hawk (in Model 1a and 1b, electronic supplementary material, figure S5), suggesting that they recognized the hawk as a predatory threat. They detected the approaching hawk and initiated an escape response later under dull light conditions and with a black background (tables 1 and 2; figure 2 and electronic supplementary material, figure S6). However, we found no significant interaction between hawk-morph and either light level or background colour (table 1 and figure 2).

Pigeons detected and reacted significantly faster if they were looking up. Furthermore, we found a significant influence of the experimental set-up, a covariate that was correlated with the speed of the approaching hawk. Pigeons reacted faster if the hawk was approaching with higher speed (table 1). Temperature had a significant effect on the reaction time in the control-hawk and light change-hawk models with pigeons reacting faster if it was warmer. It was not significantly different in the detection time (table 1). Variables time of the day, number of times the pigeon had seen the hawk or control, or hawk replicate did not have a significant influence on detection or reaction time.

## 4. Discussion

Previous studies have shown that birds avoid foraging in lower light conditions (i.e. dusk and dawn), suggesting that this is a strategy to reduce predation risk [1,10–12]. Our study provides experimental support for the hypothesis that response time will be slower under duller light conditions, thus resulting in higher predation risk in nature and providing an explanation why birds minimize foraging activity during early daylight hours [10,12]. Similarly, we found that response times were slower with a black background; further nurturing the hypothesis that birds are exposed to a higher predation risk in darker environments. Little research has been conducted on the influence of background coloration on the speed of predator detection; so far, the focus has mainly been from the perspective of the predator. For example, background complexity is an important factor contributing to an increased foraging effort [34–37]. In natural settings, the complex interplay of the light environment, background coloration, appearance and visual acuity will determine detection time and predation risk [18,38–42] and ultimately shape anti-predator behaviours with light levels and background colours being a strong selective agent in birds.

If hunting success was closely correlated to predator detection speed by prey, then our results suggest that both morphs should catch disproportionally more prey in darker light conditions, which is not the case [24]. Speed of predator detection might not be the only factor responsible for the probability of successful hunting, and avian predators' low visual acuity under darker conditions might likewise limit their own foraging success [43].

Contrary to our predictions, we found similar response times towards the two black sparrowhawk morphs irrespective of light levels or background colour. The contrasting provisioning rates in relation to light levels in the two colour morphs were proposed to be due to improved crypsis [24]. However, tracking data already cast doubt on this idea, showing that the two morphs expressed different foraging activities with dark morphs foraging more under darker conditions and light morphs equally over all light levels [25]. Thus, higher provisioning rates of dark morph individuals might simply reflect higher foraging effort and not a crypsis advantage under darker conditions. Increased foraging effort could also reflect a compensation for the hawk's own low visual acuity under darker conditions. For light morph hawks, a higher provisioning rate in brighter conditions [24] was not related to increased foraging effort [25], although our results suggest it is not explained by improved hunting success through better crypsis either, since reaction time did not differ between morphs and light levels. Thus, the mechanism for the selective advantage of the two morphs remains unidentified.

**Table 1.** Results of Models 1, 2 and 3. Effect sizes of type of trial (either towards a hawk mount/control), morph of the hawk (light/dark), light (dull/bright), background (black/white) and an interaction between light/background with the hawk-morph, and covariates speed (of hawk or control) or experiment set-up ID (experiment ID), head position of the pigeon (down at the feeder or looking up) and temperature on (a) detection and (b) reaction time of pigeons. Sample size (N) for each model given. The LMER was fitted with a log-function. The key variables for each model are indicated in bold.

| response variable | reference | (a) detection time, N = 451 | | | | (b) reaction time, N = 407 | | | |
|---|---|---|---|---|---|---|---|---|---|
| fixed-effects | category | estimate | s.e. | $\chi^2$ | Pr($>\chi^2$) | estimate | s.e. | $\chi^2$ | Pr($>\chi^2$) |
| Model 1—*control-hawk experiment* | | | | | | | | | |
| **model type** | **hawk** | **−0.060** | **0.013** | **22.332** | **<0.001** | **−0.141** | **0.018** | **61.236** | **<0.001** |
| pigeon head position | looking up | −0.185 | 0.011 | 300.146 | <0.001 | −0.170 | 0.014 | 152.14 | <0.001 |
| light | low | 0.039 | 0.008 | 22.402 | <0.001 | 0.055 | 0.011 | 25.770 | <0.001 |
| experiment ID | | 0.044 | 0.023 | 3.729 | 0.053 | 0.145 | 0.031 | 22.510 | <0.001 |
| temperature | | | | | | 0.009 | 0.004 | 5.156 | 0.023 |
| (intercept) | | 0.272 | 0.047 | 32.98 | <0.001 | −0.108 | 0.097 | 1.232 | 0.267 |

| response variable | reference | (a) detection time, N = 598 | | | | (b) reaction time, N = 554 | | | |
|---|---|---|---|---|---|---|---|---|---|
| fixed-effects | category | estimate | s.e. | $\chi^2$ | Pr($>\chi^2$) | estimate | s.e. | $\chi^2$ | Pr($>\chi^2$) |
| Model 2—*light level-hawk experiment* | | | | | | | | | |
| **light** | **dull** | **0.025** | **0.010** | **5.756** | **0.016** | **0.045** | **0.013** | **12.237** | **<0.001** |
| **hawk-morph** | **light** | **−0.001** | **0.011** | **0.003** | **0.960** | **−0.002** | **0.013** | **0.012** | **0.911** |
| pigeon head position | looking up | −0.171 | 0.010 | 321.466 | <0.001 | −0.168 | 0.012 | 192.905 | <0.001 |
| experiment ID | | 0.051 | 0.008 | 41.593 | <0.001 | 0.117 | 0.011 | 120.025 | <0.001 |
| temperature | | | | | | 0.008 | 0.003 | 6.575 | 0.01 |
| **light × hawk-morph** | **light morph × dull light** | **0.021** | **0.014** | **2.051** | **0.152** | **−0.175** | **0.070** | **1.241** | **0.265** |
| (intercept) | | 0.196 | 0.020 | 96.001 | <0.001 | −0.175 | 0.070 | 6.289 | 0.012 |

(Continued.)

**Table 1.** (*Continued.*)

| response variable | reference | (a) detection time, N = 271 | | | | (b) reaction time, N = 254 | | | |
|---|---|---|---|---|---|---|---|---|---|
| fixed-effects | category | estimate | s.e. | $\chi^2$ | Pr(>$\chi^2$) | estimate | s.e. | $\chi^2$ | Pr(>$\chi^2$) |
| *Model 3—background–hawk experiment* | | | | | | | | | |
| **background** | **white** | **−0.030** | **0.014** | **4.843** | **0.028** | **−0.009**[a] | **0.011**[a] | **8.510**[a] | **0.004**[a] |
| **hawk-morph** | **light** | **0.012** | **0.014** | **0.674** | **0.412** | **−0.005** | **0.016** | **0.086** | **0.769** |
| pigeon head position | | −0.127 | 0.015 | 74.583 | <0.001 | −0.085 | 0.018 | 22.781 | <0.001 |
| speed of the hawk | | −0.078 | 0.039 | 4.119 | 0.042 | | | | |
| **background ×** | **light morph ×** | **0.001** | **0.019** | **0.000** | **0.998** | **−0.009** | **0.022** | **0.166** | **0.684** |
| **hawk-morph** | **white** | | | | | | | | |
| | **background** | | | | | | | | |
| (intercept) | | 0.558 | 0.078 | 50.699 | <0.001 | 0.407 | 0.012 | 1063.599 | <0.001 |

aWithout interaction term.

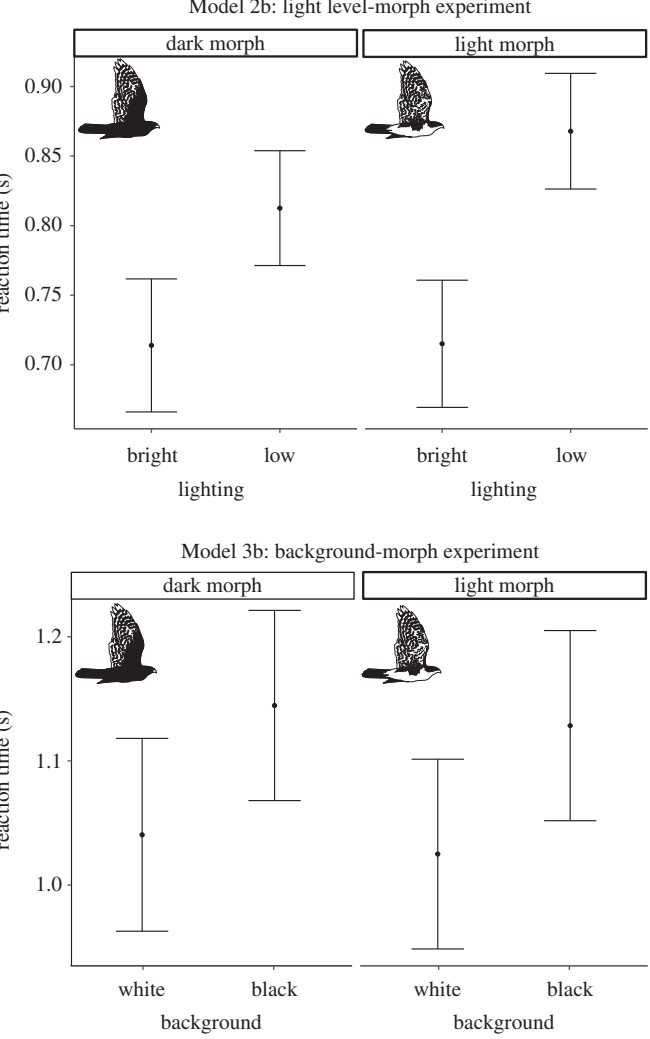

**Figure 2.** Reaction time (s) of pigeons to simulated attacks of mounts depending on light (Model 2b) or background (Model 3b). Figure based on fitted values of LMMs with 95% CIs.

**Table 2.** Least square means of response times towards a control/hawk and under varying conditions. Displayed are lsmeans ± s.e. of key explanatory variables of main results, the list of other covariates considered can be found table 1.

| | detection time (s) | | reaction time (s) | |
|---|---|---|---|---|
| | lsmeans ± s.e. | *p*-value | lsmeans ± s.e. | *p*-value |
| Model 1: *control-hawk* experiment | | | | |
| Control | 0.53 ± 0.03 s | <0.001 | 0.82 ± 0.04 s | <0.001 |
| trial (hawk mount) | 0.38 ± 0.02 s | | 0.54 ± 0.01 s | |
| Model 2: *light level-morph* experiment | | | | |
| dull light | 0.47 ± 0.02 s | 0.016 | 0.66 ± 0.02 s | <0.001 |
| bright light | 0.38 ± 0.02 s | | 0.56 ± 0.01 s | |
| Model 3: *background-morph* experiment | | | | |
| black background | 0.66 ± 0.03 s | 0.028 | 0.99 ± 0.03 s | 0.004[a] |
| white background | 0.56 ± 0.03 s | | 0.90 ± 0.03 s | |

[a]Without interaction term.

Physiological adaptation to different light or temperature conditions may be a reasonable, alternative explanation [44,45].

We did not find support that predator detectability could be a driver for the evolution of colour-polymorphism in raptors [19–21,24,25]. However, there are caveats to our experiment: crypsis works best if animals are motionless [46,47]. Thus, the crypsis advantage might only come into place when the hawk is perched (as suggested in a correlative study [48]), which our experiment would not reveal. Furthermore, we chose the position of the light source in a way that the morphs could be easily distinguished by the pigeon; however, predators might use different approach strategies in relation to the position of the sun (i.e. great-white sharks *Carcharodon carcharias* [49]). Nothing is known about morph-specific hunting techniques in the colour-polymorphic black sparrowhawk. Lastly, the light we used is artificial and the interaction of prey's vision and environmental spectral reflectance under natural conditions [50,51] might be the crucial factor in the detection of different morphs in different habitats.

Despite finding for the black sparrowhawk that crypsis was not morph-dependent, we still recommend similar experiments to be carried out in other study systems of colour-polymorphic raptors to identify the drivers of adaptive colour-polymorphism. Recently, a difference in crypsis dependent on morph and moonlight was found in the barn owl, with white owls having a foraging advantage on moonlit nights in comparison to reddish owls [52,53,54]. Similarly, a difference in crypsis under varying environmental conditions is suspected in tawny owls [22].

## 5. Conclusion

Our experiments show an effect of environmental conditions (light and background colour) on the response times of pigeons, thus providing the support that birds experience higher predation risk in darker settings and environments. We did not, however, find support for morph-dependent detectability under different visual conditions.

Ethics. This study was conducted under a CapeNature Permit (CN44–30-4175) and was approved by the UCT's SFAEC (Permit no. 2018/v5/AA).

Data accessibility. The dataset is available as electronic material on UCT's ZivaHub: doi:10.25375/uct.8332436.

Authors' contributions. A.A. and P.S. conceived the experiment. C.N. and A.P. performed the data collection—volunteers are acknowledged accordingly—and video analysis, C.N. undertook the statistical analysis. The manuscript was prepared by C.N., P.S. and A.A. and approved by all co-authors.

Competing interests. We declare we have no competing interests.

Funding. This study was funded by the DST-NRF Centre of Excellence. C.N. received funding from UCT's International Student's Scholarship and the William C. Anderson Memorial Award (RRF); and P.S. from the Claude Leon Foundation. A.P. was financially supported by the Bourse de Stage à l'Étranger (BSE).

Acknowledgements. We thank W. Cresswell and R. Thomson for advice on the experimental set-up, K. Walker and S. McCarren for their help collecting data and S.C. McPherson and A. Jenkins for catching pigeons. We thank H. Chalmers and the team of Eagle Encounters for making their resources available to us and housing the pigeons that were used in this experiment. Additional thanks go to I. Prins, whose expertise on light equipment was very helpful. Furthermore, we thank the staff of PhotoHire that offered useful information regarding light equipment and emergencies during the experimental days. We thank two anonymous reviewers for their comments that greatly improved this manuscript.

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
