## [Reviewer comments · Royal Society Open Science]

Review History

RSOS-190677.R0 (Original submission)

Review form: Reviewer 1

Is the manuscript scientifically sound in its present form?

Yes

Are the interpretations and conclusions justified by the results?

Yes

Is the language acceptable?

Yes

Is it clear how to access all supporting data?

Not Applicable

Do you have any ethical concerns with this paper?

No

Have you any concerns about statistical analyses in this paper?

No

Recommendation?

Accept as is

Comments to the Author(s)

The authors tested whether pigeons react differently when attacked by dark vs. light sparrowhawk (a colour polymorphic species) under different light conditions. They found that pigeons indeed react different if light conditions are dark or light but not in intereaction with sparrowhawk plumage coloration. This is contrary to their prediction.

I quite liked the study and recommend publication. I just wonder whether pigeons are really attacked by these sparrowhawks? And if yes, how frequently are pigeons captured by sparrowhawks? Maybe it could worth using other prey species. And I wonder whether the experimental design is appropriate given that detection time could be measured in only 9.5% of the times?

Even if the authors did not find an effect as expected, I believe that this paper is a useful contribution.

Review form: Reviewer 2

Is the manuscript scientifically sound in its present form?

No

Are the interpretations and conclusions justified by the results?

Yes

Is the language acceptable?

Yes

Is it clear how to access all supporting data?

Yes

Do you have any ethical concerns with this paper?

No

Have you any concerns about statistical analyses in this paper?

No

Recommendation?

Accept with minor revision (please list in comments)

Comments to the Author(s)

This interesting study investigates whether feral pigeons detect the dark or bright morph of the black sparrow hawk faster, under different conditions, and thus, whether earlier observed differences in hunting behavior and success of the two morphs may result from prey behavior. The study is mostly well-done and has clearly described results.

1. My main problem relates to terminology and measurements of light condition. No reader can have any idea what a "dull light condition" could mean. So here the reader needs to know the light intensity: The easiest way is to say whether this means light conditions on a cloudy day (5 to 100 times dimmer than a sunny day), in a light (10 x dimmer) or thick (up to 1000 times dimmer) forest, or maybe early or late dusk (which includes a huge range of light intensities). In addition, light measurements are useful, either in human-defined IS units (lux or Candela), but possibly even in spectral data, specifically when working indoors, where light spectra often are completely different to natural spectra. For instance, does the light have a UV component or not? Chickens have different flicker fusion frequency with and without UV, and it is not known whether the same applies to pigeons.

2. The second part of this problem relates to the description of the background and predator dummy intensities. These are not given anywhere. The most relevant information would be a measurement (if these are reasonably close to the black-grey-white colour range this could be done with a candelameter that measures the light reflected from a defined area into a defined angle (Cd/m²) of the predator dummy and the background, from the direction of the pigeon. The contrast between these two measurements is telling you whether indeed the dark morph had lower contrast against the black background and the bright morph, against the white background. From the very nice sketch of the set-up, I am not sure this really is the case. It all depends on the illumination conditions, so it needs to be measured. It is important because it allows you to answer the question whether the behavior of the pigeons did not differ, simply because contrasts were not different, or whether the contrasts really differed but the pigeons did not care.

Handheld instruments that can measure this are not very expensive and they are quite common. Terms like dark and bright are far too unspecific to be used in the context of such studies, unless they come with a measurement.

3. I am a bit disappointed that the authors missed a very similar case of colour dimorphism, in barn owls, which have a white and a dark morph as well. A lot of work has been done on that system, mostly by the group of Alexandre Roulin. A short discussion on this very similar case would bring a more general perspective to the problem and thus make this paper more interesting to general readers.

Other than these easily fixed points, I think the study is very solid.

Decision letter (RSOS-190677.R0)

25-Jun-2019

Dear Ms Nebel

On behalf of the Editors, I am pleased to inform you that your Manuscript RSOS-190677 entitled "Response time of an avian prey to a simulated hawk attack is slower in darker conditions, but is independent of hawk colour morph" has been accepted for publication in Royal Society Open Science subject to minor revision in accordance with the referee suggestions. Please find the referees' comments at the end of this email.

The reviewers and handling editors have recommended publication, but also suggest some minor

revisions to your manuscript. Therefore, I invite you to respond to the comments and revise your manuscript.

- Ethics statement

- Data accessibility

If you wish to submit your supporting data or code to Dryad (<http://datadryad.org/>), or modify your current submission to dryad, please use the following link:
<http://datadryad.org/submit?journalID=RSOS&manu=RSOS-190677>

- Competing interests

- Authors' contributions

- Acknowledgements

- Funding statement

Please ensure you have prepared your revision in accordance with the guidance at

<https://royalsociety.org/journals/authors/author-guidelines/> -- please note that we cannot publish your manuscript without the end statements. We have included a screenshot example of the end statements for reference. If you feel that a given heading is not relevant to your paper, please nevertheless include the heading and explicitly state that it is not relevant to your work.

Because the schedule for publication is very tight, it is a condition of publication that you submit the revised version of your manuscript before 04-Jul-2019. Please note that the revision deadline will expire at 00.00am on this date. If you do not think you will be able to meet this date please let me know immediately.

Please note that Royal Society Open Science charge article processing charges for all new

submissions that are accepted for publication. Charges will also apply to papers transferred to Royal Society Open Science from other Royal Society Publishing journals, as well as papers submitted as part of our collaboration with the Royal Society of Chemistry (<http://rsos.royalsocietypublishing.org/chemistry>).

on behalf of Kevin Padian (Subject Editor)
openscience@royalsociety.org

Reviewer comments to Author:

Reviewer: 1

Comments to the Author(s)

The authors tested whether pigeons react differently when attacked by dark vs. light sparrowhawk (a colour polymorphic species) under different light conditions. They found that pigeons indeed react different if light conditions are dark or light but not in interaction with sparrowhawk plumage coloration. This is contrary to their prediction.

I quite liked the study and recommend publication. I just wonder whether pigeons are really attacked by these sparrowhawks? And if yes, how frequently are pigeons captured by sparrowhawks? Maybe it could worth using other prey species. And I wonder whether the experimental design is appropriate given that detection time could be measured in only 9.5% of the times?

Even if the authors did not find an effect as expected, I believe that this paper is a useful contribution.

Reviewer: 2

Comments to the Author(s)

This interesting study investigates whether feral pigeons detect the dark or bright morph of the black sparrow hawk faster, under different conditions, and thus, whether earlier observed differences in hunting behavior and success of the two morphs may result from prey behavior. The study is mostly well-done and has clearly described results.

1. My main problem relates to terminology and measurements of light condition. No reader can

have any idea what a “dull light condition” could mean. So here the reader needs to know the light intensity: The easiest way is to say whether this means light conditions on a cloudy day (5 to 100 times dimmer than a sunny day), in a light (10 x dimmer) or thick (up to 1000 times dimmer) forest, or maybe early or late dusk (which includes a huge range of light intensities). In addition, light measurements are useful, either in human-defined IS units (lux or Candela), but possibly even in spectral data, specifically when working indoors, where light spectra often are completely different to natural spectra. For instance, does the light have a UV component or not? Chickens have different flicker fusion frequency with and without UV, and it is not known whether the same applies to pigeons.

2. The second part of this problem relates to the description of the background and predator dummy intensities. These are not given anywhere. The most relevant information would be a measurement (if these are reasonably close to the black-grey-white colour range this could be done with a candelameter that measures the light reflected from a defined area into a defined angle (Cd/m²) of the predator dummy and the background, from the direction of the pigeon. The contrast between these two measurements is telling you whether indeed the dark morph had lower contrast against the black background and the bright morph, against the white background. From the very nice sketch of the set-up, I am not sure this really is the case. It all depends on the illumination conditions, so it needs to be measured. It is important because it allows you to answer the question whether the behavior of the pigeons did not differ, simply because contrasts were not different, or whether the contrasts really differed but the pigeons did not care.

Handheld instruments that can measure this are not very expensive and they are quite common. Terms like dark and bright are far too unspecific to be used in the context of such studies, unless they come with a measurement.

3. I am a bit disappointed that the authors missed a very similar case of colour dimorphism, in barn owls, which have a white and a dark morph as well. A lot of work has been done on that system, mostly by the group of Alexandre Roulin. A short discussion on this very similar case would bring a more general perspective to the problem and thus make this paper more interesting to general readers.

Other than these easily fixed points, I think the study is very solid.

Author's Response to Decision Letter for (RSOS-190677.R0)

See Appendix A.

Decision letter (RSOS-190677.R1)

08-Jul-2019

Dear Ms Nebel,

I am pleased to inform you that your manuscript entitled "Response time of an avian prey to a simulated hawk attack is slower in darker conditions, but is independent of hawk colour morph" is now accepted for publication in Royal Society Open Science.

on behalf of Kevin Padian (Subject Editor)
openscience@royalsociety.org

Appendix A

Author responses

Reviewer comments to Author: Author's replies are in *blue italics*

Reviewer: 1

Comments to the Author(s)

The authors tested whether pigeons react differently when attacked by dark vs. light sparrowhawk (a colour polymorphic species) under different light conditions. They found that pigeons indeed react different if light conditions are dark or light but not in intereaction with sparrowhawk plumage coloration. This is contrary to their prediction.

I quite liked the study and recommend publication. I just wonder whether pigeons are really attacked by these sparrowhawks? And if yes, how frequently are pigeons captured by sparrowhawks? Maybe it could worth using other prey species.

Thank you for this question: Black sparrowhawks are almost exclusively bird hunters and their main prey items are pigeons and doves. Prey analyses in our study population have shown that about 28 % of all prey items brought to the nest are feral pigeons. We have now added some text to explain this in Material and Methods:

“Black sparrowhawks are regularly prey on feral pigeons in our study area and analysis of prey remains have shown that about 28 % of all prey items recorded are feral pigeons (Suri et al. 2017, Ibis 159:38-54) .”

And I wonder whether the experimental design is appropriate given that detection time could be measured in only 9.5% of the times?

I appears there has been some misunderstanding here.

The original text in the paper says “In 9.5% of all trials, we were unable to measure a detection time, because the pigeon was already looking directly in the direction of the on-coming hawk or did not notice the hawk.”

Thus, it is not the case that detection time was measured in only 9.5% of cases, but rather that we could NOT measure it in 9.5% of cases.

In the hope of avoiding any similar confusion in the future, we have now altered the text to read: “We were unable to measure reaction time in 9.5 % of all trials, because the pigeon was already looking directly in the direction of the on-coming hawk or did not notice it.”

Reviewer: 2

Comments to the Author(s)

This interesting study investigates whether feral pigeons detect the dark or bright morph of the black sparrow hawk faster, under different conditions, and thus, whether earlier observed differences in hunting behavior and success of the two morphs may result from prey behavior. The study is mostly well-done and has clearly described results.

1. My main problem relates to terminology and measurements of light condition. No reader can have any idea what a “dull light condition” could mean. So here the reader needs to know the light intensity: The easiest way is to say whether this means light conditions on a cloudy day (5 to 100 times dimmer than a sunny day), in a light (10 x dimmer) or thick (up to 1000 times dimmer) forest, or maybe early or late dusk (which includes a huge range of light intensities).

Thank you for raising this useful point. We have added the following sentence to make a comparison with everyday situation. This will hopefully make it easier for the reader to grasp and understand under which light conditions our experiment was carried out.

“We created two light treatments: “bright light”, using four lamps on highest intensity (2182 ± 65 lux); and “dull light” with two dimmed lamps (112 ± 12 lux, Figure 1). In comparison to a real life situation, 112 lux would be comparable to the light during a very dark overcast day, i.e. as it is encountered in a thick forest or during the early or late hours of the day. The bright light situation, whilst considerably brighter, is comparable to an overcast day during noon. This maximum light intensity was limited by the luminance output of our lamps and the heat production. Our dullest light intensity was limited by the capabilities of our camera to record interpretable images (see Supplementary Material Table S1). “

In addition, light measurements are useful, either in human-defined IS units (lux or Candela), but possibly even in spectral data, specifically when working indoors, where light spectra often are completely different to natural spectra. For instance, does the light have a UV component or not? Chickens have different flicker fusion frequency with and without UV, and it is not known whether the same applies to pigeons.

Thank you for raising this interesting point – we have now included this following text

The bulbs used were tungsten halogen incandescent lamps that produce a continuous spectrum of light, including near UV light (zeiss-campus.magnet.fsu.edu).

We also do provide information on the light intensities as SI unit (lux) for the dull and bright light level in the material and method part, see here: “We created two light treatments: “bright light”, using four lamps on highest intensity (2182 ± 65 lux); and “dull light” with two dimmed lamps (112 ± 12 lux, Figure 1).”

2. The second part of this problem relates to the description of the background and predator dummy intensities. These are not given anywhere. The most relevant information would be a measurement (if these are reasonably close to the black-grey-white colour range this could be done with a candela meter that measures the light reflected from a defined area into a defined angle (Cd/m²) of the predator dummy and the background, from the direction of the pigeon. The contrast between these two measurements is telling you whether indeed the dark morph had lower contrast against the black background and the bright morph, against the white background. From the very nice sketch of the set-up, I am not sure this really is the case. It all depends on the illumination conditions, so it needs to be measured. It is important because it allows you to answer the question whether the behavior of the pigeons did not differ, simply because contrasts were not different, or whether the contrasts really differed but the pigeons did not care.

Again, thank you for raising this interesting point.

To tackle this issue, we have measured the contrast between the hawk and the background on photos obtained directly from our video footage. The contrast ratio (relative luminance, L), based on the RGB colour space, shows that a light morph has less contrast in front of a white background and a dark morph in front of a black background. Contrary, light morphs show a higher contrast in front of a black background and dark morphs in front of a white background.

We have now added the following text to the main paper

“We confirmed that the different hawk morphs were contrasting with the background colour by calculating the contrast ratio (relative luminance). For full details see Table S2 & Figure S3-S4.”

And we provide more information (see below) to the reader on the methods and results, in the supplementary. We hope this is to the satisfaction of the reviewer.

Contrast ratio between hawk morphs and backgrounds

We used the online tool based on the Web Content Accessibility Guidelines 2.0 (see contrast-ratio and WCAG10) to calculate the contrast ratio between the background and hawk colour in the RGB colour space, defined as the relative luminance (L). The highest ratio is obtained by plain black and plain white (L = 22) whereas a minimum score is reached by the same colours (L = 1). A high ratio therefore implies high colour contrast (and good visibility) whereas a low ratio indicates low colour contrast (better crypsis).

In our background experiment, we encounter two different light-background conditions: (1) low light – white background and (2) low light – black background. In the light-change experiment, we encounter two different light-background conditions; we used (1) bright light – white background (2) low light – white background

First, we evaluated the consistency in space and time of the background colouration. It showed a high consistency where the hawk first came into view, therefore we chose one pixel of the background colouration at the beginning of every trial video as the background colour.

Second, the contrast of the hawk against the background was measured at four fixed points: first, the breast (one pixel at the front, one in the back) and, second, the underwing coverts (left and right side). These four points are representing areas of high plumage colouration differences between the morphs. Here we expect to see the differences of the contrast ratio between morphs to show a difference with light and dark morphs having a high contrast ratio against a black or white background, respectively, and low contrast values where the colour of the hawk matches the colour of the background.

Table S2 shows the mean contrast ratio (relative luminance, L) and its standard deviation of the two black sparrowhawk morphs (dark or light) against the background colouration under varying conditions during this experiment. For the light-change experiment, attacks were simulated either under bright light – white background or low light – white background. In the background-change experiment, attacks were simulated either under low light – white background or low light – black background.

Conditions	Morph	
	Light	Dark
Bright light – White background	1.35 (SD 0.33)	7.56 (SD 2.16)
Low light – White background	1.55 (SD 0.34)	2.8 (SD 0.09)
Low light – Black background	5.49 (SD 2.22)	1.20 (SD 0.10)

Figure S3 contrast ratio of the ventral colouration of the light and dark black sparrowhawk morph against the background colour in the background-change experiment. The conditions are

Low – Black: low light – black background and Low – White: low light and white background. Solid circles depict the contrast ratio (L) mean, error bars depict standard deviation. Mean and standard deviation were calculated based on four ventral point measurements per hawk (in total 20 measurements, ten per morph, four per hawk replicate).

Figure S4 contrast ratio of the ventral colouration of the light and dark black sparrowhawk morph against the background colour in the light-change experiment. The conditions are Bright – White: bright light – white background and Low – White: low light and white background. Solid circles depict the contrast ratio (L) mean, error bars depict standard deviation. Mean and standard deviation were calculated based on four ventral point measurements per hawk (in total 20 measurements, ten per morph, four per hawk replicate).

The result of the contrast ratio shows a high consistency between trials for the two colour morphs. These results validate our methodology to measure contrast in the RGB colour space.

The ventral side shows varying contrast ratios, depending on the background colour and the morph of the attacking hawk. The light morph shows a high contrast ratio when attacking in front of a black background, similarly we obtained high contrast ratio for a dark morph attacking in front of a white background. Low contrast ratio values were recorded for the dark morph attacking in front of a black background and for a light morph attacking in front of a white background (Table S2, Figure S3).

No such effect was found for the light-change experiment – where the background colour stayed the same and only the light condition was altered. The contrast ratio measurements show that the contrast is very high for the dark morph under bright light levels but evens out and becomes more similar to the contrast ratios of the light morph when the light level is decreased. No large drop of the contrast ratio is observed for the light morph, likely because the background colour and the colour of the hawk mount were similarly affected by a change of light conditions (Table S2, Figure S4).”

3. I am a bit disappointed that the authors missed a very similar case of colour dimorphism, in barn owls, which have a white and a dark morph as well. A lot of work has been done on that system, mostly by the group of Alexandre Roulin. A short discussion on this very similar case would bring a more general perspective to the problem and thus make this paper more interesting to general readers.

We have now added text to the Introduction to introduce two study systems on colour-polymorphic owls (including the barn owl system studied by A. Roulin) to the reader and added this paragraph to the introduction:

*“Barn owl females (*Tyto alba*) show different habitat use, with reddish females occupying territories with less wooded areas compared to white females. In the tawny owl (*Strix aluco*) rufous birds occupied more wooded territories than grey birds, a pattern thought to be driven by crypsis advantages for the different morphs [22].”*

and to the discussion:

“Despite finding for the black sparrowhawk that crypsis was neither morph- nor environmentally-dependent, we still recommend similar experiments to be carried out in other study systems of colour-polymorphic raptors to identify the drivers of adaptive colour-polymorphism. For both barn and tawny owls, a difference in crypsis under varying environmental conditions is suspected [22, 52, 53], but has not been experimentally tested yet.”

Other than these easily fixed points, I think the study is very solid.

Thank you for your useful comments.